# Reinforcement Fine-Tuning Naturally Mitigates Forgetting in Continual Post-Training

Song Lai [* 1 2]   Haohan Zhao [* 1 2]   Rong Feng [1 2]   Changyi Ma [3]   Wenzhuo Liu [4 5]   Hongbo Zhao [4 5]
Xi Lin [6]   Dong Yi [1]   Qingfu Zhang [2]   Hongbin Liu [1]   Gaofeng Meng [1 4 5]   Fei Zhu [1]

## Abstract

Continual post-training (CPT) is a popular and effective technique for adapting foundation models like multimodal large language models to ever-evolving downstream tasks. While existing research primarily focuses on methods like data replay, model expansion, or parameter regularization, the fundamental role of the learning paradigm remains largely unexplored. This paper presents a comparative analysis of two core post-training paradigms: supervised fine-tuning (SFT) and reinforcement fine-tuning (RFT), investigating their respective impacts on knowledge retention during CPT. Our experiments are conducted across multiple multimodal tasks, utilizing Qwen2.5-VL-7B-Instruct as the base model. The investigation yields two significant findings: (1) When continuously learning on downstream tasks, SFT leads to catastrophic forgetting of previously learned tasks. In contrast, RFT inherently preserves prior knowledge and achieves performance comparable to multi-task training. (2) RFT successfully protects and even enhances the model's general knowledge on standard benchmarks, while SFT degrades general model capabilities severely. Further analysis reveals that this stability is not primarily due to explicit mechanisms like KL penalty or chain-of-thought reasoning. We investigate RFT's learning dynamics and find that its selective update mechanism inherently

prevents interference with established knowledge. Based on this insight, we propose a rollout-based instance filtering algorithm (RIF-RFT) that enhances the training efficiency of RFT by focusing on learnable samples. Our comprehensive study demonstrates the superiority of RFT as a robust paradigm for continual post-training.

## 1. Introduction

Recent advancements in multimodal large language models (MLLMs) have demonstrated remarkable capabilities in complex world understanding (Achiam et al., 2023; Liu et al., 2024; Wang et al., 2024a). To align with the demands of real-world deployment, MLLMs must adapt to a stream of data and evolving user requirements, incorporating new skills and domain knowledge over time (Zhu et al., 2024). This calls for an efficient and scalable continual post-training (CPT) paradigm. A key challenge in CPT is the well-known phenomenon of catastrophic forgetting (McCloskey & Cohen, 1989), where adapting to a new task leads to a severe degradation of performance on previously learned tasks. To reduce forgetting, recent studies (Guo et al., 2025c) focus on data replay (Maharana et al., 2025; Lee et al., 2025; Wang et al., 2025), model expansion (Zhao et al., 2025; Guo et al., 2025b; Zeng et al., 2025), and explicit regularization (Liu et al., 2025a). Nevertheless, existing methods typically leverage the supervised fine-tuning (SFT) paradigm by default, and the role of the fundamental fine-tuning paradigm in CPT has been overlooked.

Recently, reinforcement fine-tuning (RFT), which optimizes models based on feedback from generated outputs, has significantly advanced foundation model post-training (Chu et al., 2025; Shao et al., 2024; Guo et al., 2025a). To the best of our knowledge, this work presents the first direct comparative investigation into whether SFT or RFT is the more suitable paradigm for CPT, focusing on knowledge preservation for both specific downstream tasks and general capabilities. Experimentally, we continually fine-tune the Qwen2.5-VL-7B-Instruct model (Bai et al., 2025) on a benchmark comprising diverse multimodal tasks covering

---
[*]Equal contribution   [1]Centre for Artificial Intelligence and Robotics, Hong Kong Institute of Science & Innovation, Chinese Academy of Sciences, Hong Kong, China [2]City University of Hong Kong, Hong Kong, China [3]School of Artificial Intelligence, Jilin University, Changchun, China [4]Institute of Automation, Chinese Academy of Sciences, Beijing, China [5]University of Chinese Academy of Sciences, Beijing, China [6]School of Mathematics and Statistics, Xi'an Jiaotong University, Xi'an, China. Correspondence to: Fei Zhu <fei.zhu@cair-cas.org.hk>, Gaofeng Meng <gaofeng.meng@cair-cas.org.hk>.

*Proceedings of the 43rd International Conference on Machine Learning*, Seoul, South Korea. PMLR 306, 2026. Copyright 2026 by the author(s).

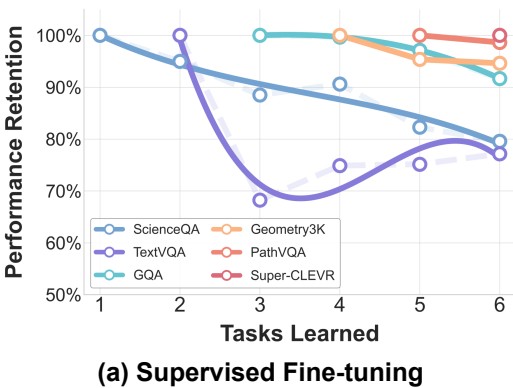

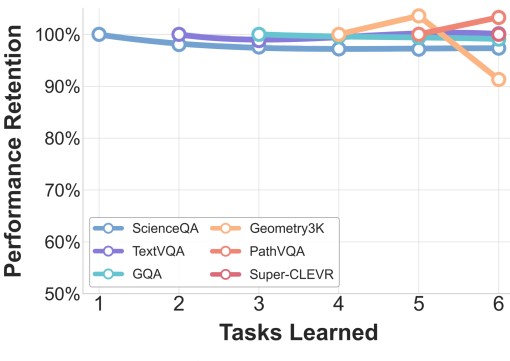

**(a) Supervised Fine-tuning**                    **(b) Reinforcement Fine-tuning**

*Figure 1.* Comparison of performance retention between SFT and RFT in continual post-training. We plot the performance on each task, normalized relative to its initial post-training peak, as the model learns through a sequence of multimodal tasks. **(a)** SFT exhibits classic catastrophic forgetting, where performance on previously learned tasks degrades dramatically as new tasks are introduced. **(b)** By contrast, RFT demonstrates remarkable stability, maintaining high performance on prior tasks throughout the entire sequence. This suggests an inherent forgetting-mitigation property within the RFT paradigm. Further details on the experimental setup can be found in Section 4.

various domains. To fully reflect the knowledge preservation ability, we evaluate forgetting on both learned specific tasks and general benchmarks such as MMMU (Yue et al., 2024), MMLU-Pro (Wang et al., 2024b), and POPE (Li et al., 2023a).

The empirical investigation yields two notable findings: **(1)** As shown in Figure 1, when continuously learning on downstream tasks, SFT leads to catastrophic forgetting of previously learned tasks, which is consistent with existing studies (Guo et al., 2025c). In contrast, RFT can inherently protect prior knowledge of sequential post-training, maintaining strong performance on old tasks after being adapted to new tasks. Surprisingly, without any data replay, continual post-training with RFT can achieve comparable performance with that of multi-task training, which is not achievable even when equipping SFT with continual learning strategies. **(2)** As demonstrated in Figure 2, continual training on downstream tasks with SFT severely degrades general model capabilities, which is known as base model degradation (Liu et al., 2025a). For example, the performance drops from 52.1% to 40.1% on MMMU. Fortunately, RFT protects the general performance and enhances the base model's pre-existing general knowledge (52.1% → 54.2%). These observations highlight the knowledge preservation capability of RFT.

To understand how RFT mitigates forgetting during CPT, we conduct additional experiments with the popular and representative group relative policy optimization (GRPO) framework (Shao et al., 2024). We analyze the impact of KL divergence penalty and chain-of-thought (CoT) reasoning (Wei et al., 2022) on forgetting mitigation. Particularly, the KL divergence penalty prevents the policy from changing too drastically, similar to the well-known knowledge distillation in continual learning (Li & Hoiem, 2017). However, our

analysis indicates that these explicit mechanisms are not the primary drivers of forgetting mitigation. Instead, through systematic analysis of learning dynamics across samples of varying difficulty, we identify that RFT's advantage stems from its selective learning mechanism—it naturally focuses gradient updates on samples where the model can produce meaningful responses, while samples beyond current model capability contribute minimal harmful updates. Thus, we introduce a rollout-based instance filtering algorithm that enhances the stability of GRPO while still being an excellent knowledge protector.

Our main contributions are threefold:

1. We present the first comprehensive analysis of the forgetting mitigation effects of SFT and RFT during continual post-training of MLLMs, demonstrating that RFT naturally preserves not only the performance of learned downstream tasks but also general model capabilities.

2. Through comprehensive ablation studies and empirical analysis of learning dynamics, we demonstrate that RFT's forgetting mitigation is not attributable to KL regularization or CoT reasoning. We provide empirical evidence that RFT naturally concentrates learning signals on samples within model competence, offering insight into its stability.

3. We propose a rollout-based instance filtering algorithm (RIF-RFT) that enhances the training efficiency of RFT.

## 2. Related Works

**Continual Post-Training in MLLMs.** Continual learning aims to enable models to learn from a stream of tasks with-

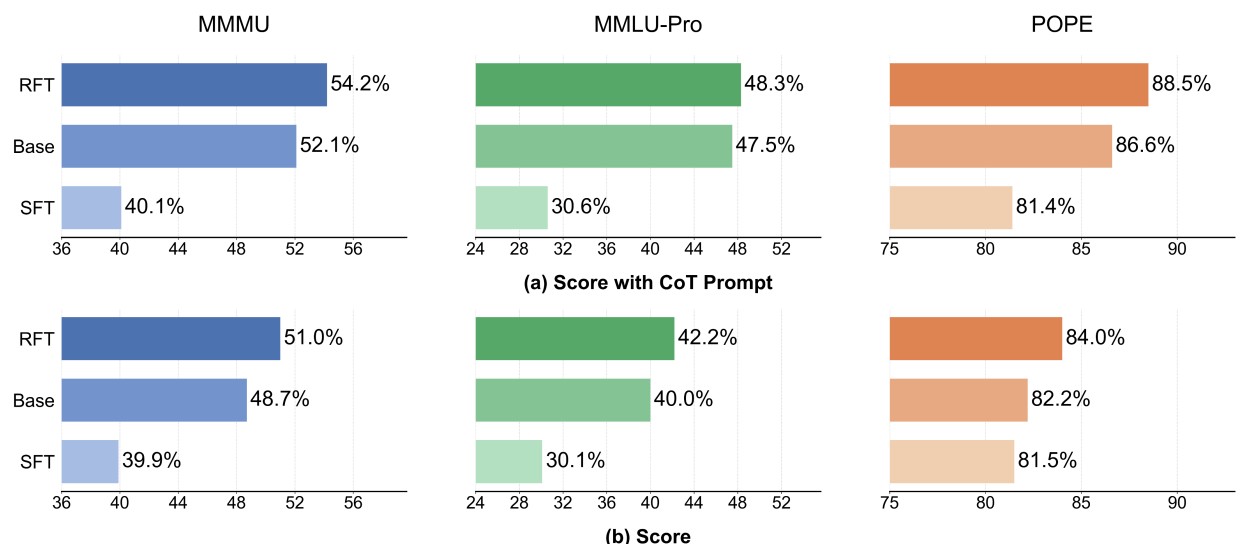

*Figure 2.* General capability preservation after continual post-training. We evaluate models at the end of learning all downstream tasks on general benchmarks using both CoT and direct prompting. Compared to the base model, SFT (shown in light colors) causes degradation while RFT (shown in darker colors) preserves and even enhances general capabilities.

out catastrophically forgetting previously acquired knowledge (Van de Ven et al., 2022). For MLLMs, this capability is particularly important for adapting these powerful models to a diverse range of downstream multimodal tasks. Existing CPT research in MLLMs focuses on adapting traditional forgetting mitigation strategies within an SFT paradigm, with benchmarks covering domain and ability continual learning (Guo et al., 2025c; Zhao et al., 2025). In addition to these methods, recent efforts to mitigate catastrophic forgetting in MLLMs primarily focus on parameter-efficient learning and dynamic data selection. For instance, HiDe-LLaVA (Guo et al., 2025b) employs a hierarchical decoupling framework for task-specific LoRA expansion and general knowledge fusion. MRLoRA (Zhao et al., 2025) leverages architectural decoupling and a multimodal routing mechanism to selectively activate specialized parameters. In terms of data management, Adapt-∞ (Maharana et al., 2025) dynamically selects high-impact samples based on gradient representations and prunes redundant data. These diverse strategies collectively aim to enhance the ability of MLLMs to continually learn new tasks while preserving previously acquired knowledge. Recently, (Liu et al., 2025a) developed LLaVA-c, which is a simple yet effective CPT framework for MLLMs, addressing task balancing and catastrophic forgetting through spectral-aware consolidation and unsupervised inquiry regularization.

**Post-Training of Foundation Models.** Post-training is a critical stage for refining the capabilities of pre-trained foundation models (Shao et al., 2024; Chu et al., 2025; Achiam et al., 2023). SFT on task-specific or instruction-formatted datasets is a common approach to adapt models to down-

stream applications (Chung et al., 2024; Zhou et al., 2023). For example, (Chung et al., 2024) demonstrated that by scaling the number of tasks and model size, and incorporating CoT data, SFT significantly enhances the performance and generalization of various large language models across diverse benchmarks. Recently, RFT has gained prominence for aligning models with human preferences or improving performance on specific objectives (Liu et al., 2025c; Zhai et al., 2024; Shao et al., 2024; Luong et al., 2024; Li et al., 2025; 2023c; Ahmadian et al., 2024). Particularly, GRPO (Shao et al., 2024) largely enhances mathematical reasoning and optimizes memory usage, being a popular method for post-training of large language models. (Liu et al., 2025b) revealed inherent biases in the GRPO algorithm, then introduces an unbiased optimization method that improves token efficiency while maintaining reasoning performance. Visual-RFT (Liu et al., 2025c) boosts MLLMs by using reinforcement learning with rule-based visual rewards, making them more data-efficient and better at various visual tasks than traditional SFT. Recently, (Chu et al., 2025) demonstrated that reinforcement learning significantly enhances the generalization capabilities of foundation models, while SFT primarily leads to memorization. In this work, we study the comparative effect of SFT and RFT on knowledge retention in MLLMs continual post-training. Some concurrent studies have also observed RFT's ability to mitigate forgetting, attributing this stability to implicit KL minimization (Shenfeld et al., 2025), the mode-seeking nature of on-policy data (Chen et al., 2025). Our work distinguishes by systematically investigating this phenomenon in the more challenging *continual* post-training setting for MLLMs, rather than single-task adaptation. Recent work by (Zhang et al.,

2025) investigates SFT and RFT from a data perspective, showing that incorporating reasoning trajectories in SFT can reduce forgetting. Their findings complement our work by highlighting how data format affects SFT's stability, while we demonstrate that RFT provides inherent forgetting mitigation without reasoning format. Together, these studies provide comprehensive guidance for post-training paradigm selection.

# 3. Preliminaries

Post-training is a critical phase following large-scale pre-training that adapts foundation models to specific downstream tasks or align them with human preferences (Ouyang et al., 2022; Kumar et al., 2025). We model the MLLM with parameters $\theta$ as a policy $\pi_\theta$. This policy defines a conditional probability distribution $\pi_\theta(a|x)$ over possible text responses $a$ given a multimodal input prompt $x$, which consists of text and images. We also assume a scalar reward function $r(x, a) \in \mathbb{R}$ that evaluates the quality of a response. Post-training aims to update the parameters $\theta$ of a pre-trained base model $\pi_{\theta_{\text{base}}}$ to improve its performance on a downstream task using a training dataset $\mathcal{D}$, which can be achieved by SFT (Ouyang et al., 2022) or RFT (Lee et al., 2023).

**SFT.** Given training dataset $\mathcal{D} = \{(x_i, a_i^*)\}_{i=1}^N$ consisting of prompts $x_i$ and their corresponding ground-truth responses $a_i^*$, SFT maximizes the likelihood of generating the ground-truth responses. This is typically achieved by minimizing the negative log-likelihood loss:

$$
\begin{aligned}
\mathcal{L}_{\text{SFT}}(\theta) &= -\mathbb{E}_{(x, a^*)\sim\mathcal{D}}[\log \pi_\theta(a^*|x)] \\
&= -\mathbb{E}_{(x, a^*)\sim\mathcal{D}}\left[\sum_{t=1}^{|a^*|} \log \pi_\theta(a_t^*|x, a_{<t}^*)\right].
\end{aligned}
\tag{1}
$$

**RFT.** In RFT, the model $\pi_\theta$ is treated as a policy, and generates one or more candidate responses for a given prompt $x$. The optimization objective is to maximize the expected reward:

$$
\mathcal{J}_{\text{RFT}}(\theta) = \mathbb{E}_{x\sim\mathcal{D}}\mathbb{E}_{a\sim\pi_\theta(\cdot|x)}[r(x, a)].
\tag{2}
$$

The gradient of this objective is typically estimated using policy gradient methods. The most basic form is the REINFORCE (Williams, 1992) estimator, which, unfortunately, has high gradient variance. Recent RFT algorithms (Shao et al., 2024; Li et al., 2023c; Ahmadian et al., 2024) address this issue by designing more stable advantage estimators and baselines. We introduce some of the representative methods used in our study below.

For a prompt $x$, **GRPO** (Shao et al., 2024) generates a group of $n$ responses $\{a_1, \ldots, a_n\}$ and computes their rewards $\{r_1, \ldots, r_n\}$. The advantage for a response $a_i$ is its normalized reward relative to the group mean: $A(a_i) = (r_i - \bar{r})/\sigma_r$, where $\bar{r}$ and $\sigma_r$ are the mean and standard deviation of the rewards. The objective is to maximize the expected advantage-weighted log-probability, often with a KL-divergence penalty against a reference policy $\pi_{\text{ref}}$ to stabilize training:

$$
\begin{aligned}
\mathcal{J}_{\text{GRPO}}(\theta) = \mathbb{E}_{x, \{a_i\}}\Bigg[ & \sum_{i=1}^n A(a_i) \log \pi_\theta(a_i|x) \Bigg] \\
& -\beta D_{\text{KL}}(\pi_\theta(\cdot|x) || \pi_{\text{ref}}(\cdot|x)),
\end{aligned}
\tag{3}
$$

where $\beta > 0$. **ReMax** (Li et al., 2023c) use the reward of a greedy decoding response $\hat{a}$ as a baseline. For a single sampled response $a$, the objective is to maximize:

$$
\mathcal{J}_{\text{ReMax}}(\theta) = \mathbb{E}_{x, a\sim\pi_\theta} \left[(r(x, a) - r(x, \hat{a})) \log \pi_\theta(a|x)\right].
\tag{4}
$$

This adaptive baseline helps to normalize rewards and reduce gradient variance. To further reduce variance, **RLOO** (Ahmadian et al., 2024) generates $n$ samples $\{a_1, \ldots, a_n\}$ and uses the average reward of the other $n-1$ samples as a baseline for sample $a_i$:

$$
\begin{aligned}
\mathcal{J}_{\text{RLOO}}(\theta) = \mathbb{E}_{x, \{a_i\}}\Bigg[ \frac{1}{n}\sum_{i=1}^n \Bigg( & r(x, a_i) \\
& -\frac{1}{n-1}\sum_{j\neq i} r(x, a_j) \Bigg) \log \pi_\theta(a_i|x)\Bigg].
\end{aligned}
\tag{5}
$$

**Continual Post-Training Formulation.** In CPT, the model learns from a sequence of $T$ tasks with datasets $\{\mathcal{D}_1, \ldots, \mathcal{D}_T\}$. The core challenge is catastrophic forgetting, i.e., a significant drop in performance on previously learned tasks. Following the general continual learning framework, CPT can be formulated as a constrained optimization problem. When learning task $t$, the objective is:

$$
\begin{aligned}
\theta^t = \arg\min_\theta \mathcal{L}(\theta; \mathcal{D}_t) \quad &\text{s.t. } \mathcal{L}(\theta; \mathcal{D}_i) \\
&\leq \mathcal{L}(\theta^i; \mathcal{D}_i), \quad \forall i \in [1, t-1]
\end{aligned}
\tag{6}
$$

where $\mathcal{L}(\theta; \mathcal{D}_i)$ is the training objective (e.g., negative log-likelihood for SFT or negative expected reward for RFT) on task $i$, and $\theta^i$ are parameters after learning task $i$.

# 4. Reinforcement Fine-Tuning Mitigates Forgetting in CPT

This section presents our comparative results comparing RFT and SFT in a continual post-training scenario. We detail our experimental setup and then present the main findings that highlight the superiority of RFT for knowledge preservation.

## 4.1. Experimental Setup

**Continual Post-Training Model & Datasets.** We adopt the open-source Qwen2.5-VL-7B-Instruct (Bai et al., 2025) as our base model, primarily due to its demonstrated superiority in vision-language comprehension and its favorable resource footprint, which is crucial for practical deployment. We continually fine-tune the model on diverse vision-language datasets (ScienceQA (Lu et al., 2022), TextVQA (Singh et al., 2019), VizWiz (Gurari et al., 2018), GQA (Hudson & Manning, 2019), Geometry3K (Lu et al., 2021), PathVQA (He et al., 2020), Super-CLEVR (Li et al., 2023b)), covering a wide range of common downstream applications. After the end of CPT, evaluation is performed on the test sets of all previously encountered tasks. Additionally, to fully assess the knowledge preservation ability, we evaluate the model on diverse, general benchmarks at the end of learning all downstream tasks. Specifically, we evaluate the model on three specialized benchmarks: MMMU (Yue et al., 2024), MMLU-Pro (Wang et al., 2024b), and POPE (Li et al., 2023a). Particularly, we include POPE to systematically assess whether CPT induces object hallucination in MLLMs. A detailed description of those datasets is provided in the Appendix A.

**Learning Algorithms & Reward.** Our experiments encompass a range of fine-tuning algorithms, including standard SFT (Zheng et al., 2024) and several representative RFT algorithms, i.e., GRPO (Shao et al., 2024), ReMax (Li et al., 2023c), and RLOO (Ahmadian et al., 2024). For both SFT and RFT, model outputs are normalized by disregarding extraneous whitespace (e.g., spaces, indentations, newlines) and ignoring case sensitivity to ensure precise assessment. For GRPO, the overall reward $r_{\text{overall}}$ is designed with a weighted sum of accuracy reward and format reward:

$$r_{\text{overall}} = 0.9r_{\text{acc}} + 0.1r_{\text{format}}. \quad (7)$$

Specifically, the accuracy reward $r_{\text{acc}}$ assesses the semantic correctness of the generated content, which yields a reward of 1 if the generated answer $a$ matches the ground truth answer $a^*$, and 0 otherwise. The format reward assesses adherence to the expected output structure. It utilized regular expressions to verify the correct presence and formatting of the CoT reasoning block, delineated by `<think>` and `</think>` tags, and the final answer encapsulated within a `\boxed{}` environment. A perfect format match resulted in a score of 1, otherwise 0.

**Prompt Template.** Our base model, Qwen-VL-7B-Instruct, utilizes two kinds of input prompt templates, as illustrated in the Appendix. *NoCoT* (non-chain-of-thought) prompt template adheres to a basic question-answering format, where the question text is presented directly, and the model is expected to provide the final answer without intermediate steps. Differently, in *CoT* prompt template, the query's question text is directly incorporated into the prompt, followed by an instruction for the model to first engage in a reasoning process. This CoT reasoning is then generated within a dedicated `<think>` and `</think>` block. The final answer is explicitly distinguished and encapsulated within a `\boxed{}` environment.

**Evaluation Metrics.** To quantify the model's performance during CPT, we adopt two standard metrics. Let $P_{t,j}$ denote the test accuracy on task $j$ after learning task $t$. We measure the final overall performance using ***average accuracy*** (***AvgAcc***), which is the average accuracy across all tasks after training on the final task $T$. To measure knowledge retention, we use the ***forgetting measure*** (***FM***), which calculates the average difference between the final accuracy of a task and the best accuracy achieved for that task throughout the training sequence. Let $P_i^* = \max_{k \in \{i,...,T\}} P_{k,i}$ be the best performance for task $i$. The above two metrics are defined as:

$$AvgAcc = \frac{1}{T}\sum_{i=1}^{T} P_{T,i}, \qquad FM = \frac{1}{T}\sum_{i=1}^{T}(P_{T,i} - P_i^*). \quad (8)$$

A higher *AvgAcc* indicates better overall performance, while an *FM* closer to zero signifies less forgetting and better knowledge preservation.

**Implementation Details.** All experiments employ full-parameter fine-tuning for both SFT and RFT to ensure comprehensive capability assessment. Experiments of SFT are conducted using the *llamafactory* (Zheng et al., 2024) framework, with a learning rate of $1e-5$ and a batch size of 24. RFT methods (GRPO, ReMax, and RLOO) are implemented using the *easyR1* (Zheng et al., 2025) framework, building upon *Verl* (Sheng et al., 2024). A consistent configuration is applied across RFT methods to ensure an equitable comparison: a learning rate of $1e-6$, a rollout batch size of 512, a sampling temperature of 1.0, with KL-divergence coefficient $\beta = 0.01$. Specifically, GRPO is implemented adhering to its foundational methodology, with a group size set to 8. ReMax followed its core algorithm, and RLOO adopted the official Hugging Face algorithm. To ensure the generality of our findings, we conduct additional experiments across different model architectures, scales, and task domains, with detailed results provided in Appendix B.

## 4.2. Finding 1: RFT Inherently Resists Catastrophic Forgetting

Our primary investigation focuses on the knowledge retention capabilities of SFT and RFT within a continual learning sequence. The results, summarized in Table 1, reveal a contrast between the two paradigms.

**SFT suffers from catastrophic forgetting.** We observe that sequential SFT leads to a severe degradation of performance on previously learned tasks with a forgetting measure (FM) of **-10.4%**. For instance, performance on ScienceQA drops dramatically (95.2% → 76.1%) after completing the entire task sequence.

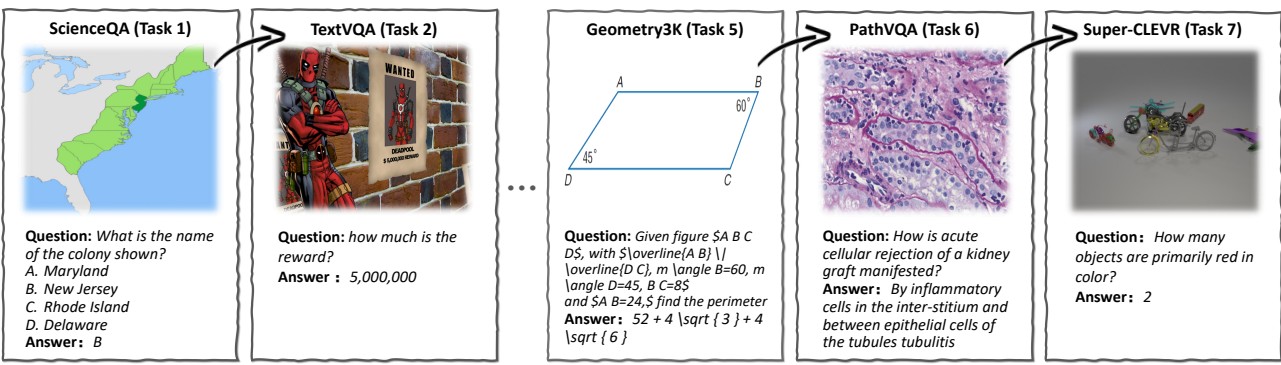

*Figure 3.* Illustrative examples of continual post-training benchmark.

*Table 1.* Final performance comparison on all tasks after the entire continual learning sequence. The **best** and second-best results are highlighted. "-" indicates that the metric is not applicable.

| Method | SciQA | TextVQA | VizWiz | GQA | Geo. | PathVQA | sCLEVR | AvgAcc | FM |
|---|---|---|---|---|---|---|---|---|---|
| Base | 90.5 | 62.8 | 45.5 | 47.2 | 37.7 | 21.8 | 41.1 | 49.5 | - |
| MTL (SFT) | 95.2 | 69.9 | 64.5 | 63.4 | 18.1 | 61.6 | 57.5 | 62.9 | - |
| SFT | 76.1 | 55.8 | 46.8 | 58.5 | 20.2 | **62.2** | **58.2** | 54.0 | -10.4 |
| ReMax | 87.6 | 71.4 | 51.6 | 62.4 | 16.8 | 33.3 | 54.1 | 53.9 | -3.8 |
| RLOO | **94.0** | 73.7 | 48.9 | 62.7 | **42.1** | 40.5 | 55.3 | 59.6 | **-2.1** |
| GRPO | 93.0 | **74.8** | **51.8** | 65.9 | 38.4 | 41.3 | 54.2 | **60.0** | -2.3 |

*Table 2.* General capabilities evaluation on MMMU, MMLU-Pro, and POPE benchmarks after training on downstream tasks. The **best** and second-best results are highlighted.

| Benchmark | Eval CoT | Base | SFT | | RFT | | |
|---|---|---|---|---|---|---|---|
| | | | SFT | MTL (SFT) | GRPO | RLOO | ReMax |
| MMMU | ✔ | 52.1 | 40.1 (↓**12.0**) | 47.8 (↓**4.3**) | **54.2** (↑**2.1**) | 53.7 (↑**1.6**) | 48.7 (↓**3.4**) |
| | ✘ | 48.7 | 39.9 (↓**8.8**) | 48.1 (↓**0.6**) | 51.0 (↑**2.3**) | 46.8 (↓**1.9**) | **51.6** (↑**2.9**) |
| MMLU-Pro | ✔ | 47.5 | 30.6 (↓**16.9**) | 33.2 (↓**14.3**) | **48.3** (↑**0.8**) | 45.1 (↓**2.4**) | 35.4 (↓**12.1**) |
| | ✘ | 40.0 | 30.1 (↓**9.9**) | 32.9 (↓**7.1**) | **42.2** (↑**2.2**) | 39.7 (↓**0.3**) | 41.0 (↑**1.0**) |
| POPE | ✔ | 86.6 | 81.4 (↓**5.2**) | 84.9 (↓**1.7**) | **88.5** (↑**1.9**) | 88.2 (↑**1.6**) | 85.2 (↓**1.4**) |
| | ✘ | 82.2 | 81.5 (↓**0.7**) | 84.5 (↑**2.3**) | 84.0 (↑**1.8**) | 82.0 (↓**0.2**) | **87.2** (↑**5.0**) |

The final average accuracy (AvgAcc) of **54.0%** is substantially lower than the multi-task learning of SFT, which is the upper bound of **62.9%**, confirming that SFT is highly susceptible to forgetting.

**RFT preserves task knowledge and achieves MTL performance.** In contrast, all RFT methods demonstrate remarkable resilience against forgetting. As shown in Table 1, RFT methods exhibit very low forgetting measures, with GRPO achieving an FM of **-2.3%**. For example, GRPO maintains ScienceQA performance at **93.0%** after learning all tasks, compared to its peak performance of **95.6%**, which is a minimal drop compared to SFT. Among RFT methods, GRPO performs best, achieving a final AvgAcc of **60.0%**, which is close to the upper bound of **62.9%**. The model achieves this high performance *without any explicit continual learning strategies*, suggesting that the RFT paradigm is inherently robust for CPT.

**RFT outperforms Experience Replay** To compare RFT's performance against established CL techniques, we compare it

with Experience Replay (ER) (Schaul et al., 2015), widely considered one of the most effective baselines (Lai et al., 2025). We implement ER with a 25% replay ratio, which represents the upper range suggested by recent work on LLM continual post-training (Abbes et al., 2025). As detailed in Table 5, while ER improves upon vanilla SFT (FM improves from -4.4% to -2.8%), it still lags behind RFT (-0.4%). Furthermore, ER introduces significant storage overhead and potential negative transfer, whereas RFT achieves superior stability inherently without requiring external memory buffers.

**Generalization and robustness analysis.** We further examine whether the forgetting-mitigation effect of RFT is robust across model scales, task domains, and task orders. First, we evaluate Qwen2.5-VL-3B-Instruct (Bai et al., 2025) and Qwen3-VL-8B-Instruct (Yang et al., 2025) on a subset of our benchmark tasks. As shown in Table 6, RFT maintains near-zero forgetting across both scales, while SFT suffers substantially larger performance drops. Second, we evaluate the text-only Qwen2.5-7B-Instruct on GSM8K (Cobbe et al., 2021) and USMLE (Jin et al., 2020).

*Table 3.* Downstream task performance for ablation models. We investigate the role of the KL term and CoT through variants of GRPO. [†] indicates that the training process is unstable and requires multiple restarts from a previous checkpoint to achieve convergence.

| Method | SciQA | TextVQA | VizWiz | GQA | Geo. | PathVQA | sCLEVR | AvgAcc |
|---|---|---|---|---|---|---|---|---|
| SFT | 76.1 | 55.8 | 46.8 | 58.5 | 20.2 | **62.2** | **58.2** | 54.0 |
| GRPO | 93.0 | 74.8 | 51.8 | **65.9** | **38.4** | 41.3 | 54.2 | **60.0** |
| GRPO w/o KL | 93.0 | **75.0** | 51.6 | **65.9** | 35.6[†] | 40.9[†] | 54.7[†] | 59.5 |
| GRPO w/o CoT | **94.7** | 74.7 | **63.8** | **65.9** | 23.8 | 38.2 | 54.4 | 59.4 |

*Table 4.* General capabilities evaluation for ablation models. Each benchmark is evaluated with and without CoT prompts to provide a comprehensive view.

| Benchmark | Eval CoT | Base | GRPO | GRPO w/o CoT | GRPO w/o KL |
|---|---|---|---|---|---|
| MMMU | ✔ | 52.1 | 54.2 (↑**2.1**) | 51.8 (↓**0.3**) | 52.2 (↑**0.1**) |
| | ✗ | 48.7 | 51.0 (↑**2.3**) | 51.6 (↑**2.9**) | 49.2 (↑**0.5**) |
| MMLU-Pro | ✔ | 47.5 | 48.3 (↑**0.8**) | 48.9 (↑**1.4**) | 45.2 (↓**2.3**) |
| | ✗ | 40.0 | 42.2 (↑**2.2**) | 41.9 (↑**1.9**) | 42.3 (↑**2.3**) |
| POPE | ✔ | 86.6 | 88.5 (↑**1.9**) | 85.3 (↓**1.3**) | 74.2 (↓**12.4**) |
| | ✗ | 82.2 | 84.0 (↑**1.8**) | 88.7 (↑**6.5**) | 87.6 (↑**5.4**) |

*Table 5.* Comparison between RFT, SFT, and SFT with ER on Qwen2.5-VL-3B-Instruct.

| Method | sCLEVR | SciQA | TextVQA | AvgAcc | FM |
|---|---|---|---|---|---|
| SFT | 51.5 | 92.3 | 67.6 | 70.5 | -4.4 |
| SFT + ER (25%) | 53.2 | 92.1 | 64.5 | 69.9 | -2.8 |
| GRPO | 57.8 | 92.7 | 72.8 | **74.4** | **-0.4** |

*Table 6.* Performance comparison across different model scales.

| Model Size | Method | sCLEVR | SciQA | TextVQA | AvgAcc | FM |
|---|---|---|---|---|---|---|
| 3B | GRPO | 57.8 | 92.7 | 72.8 | 74.4 | -0.4 |
| | SFT | 51.5 | 92.3 | 67.6 | 70.5 | -4.4 |
| 8B | GRPO | 57.0 | 96.3 | 76.1 | 76.5 | -0.2 |
| | SFT | 48.2 | 91.5 | 68.9 | 69.5 | -7.1 |

Table 7 shows that the same trend also holds beyond multimodal tasks. Third, we test two task orderings on both 3B and 8B models. Table 8 shows that RFT remains stable under different task orders, with the forgetting measure consistently close to zero.

### 4.3. Finding 2: RFT Protects and Enhances General Capabilities

Beyond task-specific knowledge, an ideal CPT process also requires preserving the model's foundational, general-purpose abilities. We evaluated the models on general benchmarks to measure this effect. The results, presented in Table 2, highlight another critical advantage of RFT.

**SFT harms general capabilities in both CL and MTL.** Our experiments reveal that SFT causes significant *base model degradation* (Liu et al., 2025a). SFT induces a severe performance drop of ↓**16.9%** on the challenging MMLU-Pro benchmark (47.5% → 30.6%). Crucially, this is not merely an artifact of sequential learning; even multi-task SFT (MTL (SFT)), which trains on all data simultaneously, still causes a severe drop of ↓**14.3%** on the same benchmark. A similar trend is evident on MMMU, where SFT and MTL (SFT) cause performance to decline by ↓**12.0%**

and ↓**4.3%** respectively. This demonstrates that the SFT paradigm itself appears harmful to the model's foundational capabilities.

**RFT preserves and enhances general capabilities.** In contrast to the capability decay observed across all SFT methods, the RFT paradigm effectively safeguards the model's general abilities. GRPO, in particular, often *enhances* these abilities. For instance, GRPO improves performance on MMMU by ↑**2.1%** (52.1% → 54.2%). Crucially, RFT also improves model general capabilities, with GRPO improving the POPE score by ↑**1.9%** (86.6% → 88.5%) and reducing the tendency for hallucination. This clear difference highlights that RFT is a more robust paradigm for continual post-training. Additional POPE split-wise results are provided in Appendix B.4, where the same trend holds across random, popular, and adversarial splits.

## 5. Understanding RFT's Forgetting Mitigation

To investigate the mechanisms behind RFT's remarkable stability, this section presents a series of ablation studies based on the popular and representative GRPO algorithm (Shao et al., 2024).

### 5.1. The Roles of CoT and KL Penalty

We test two primary hypotheses: (**1**) The KL-divergence penalty, by regularizing policy updates, acts as a form of knowledge distillation (Li & Hoiem, 2017) that preserves past knowledge. (**2**) The complex reasoning structure of CoT builds more abstract and resilient knowledge representations, protecting them from being overwritten. Thus, we evaluate three GRPO variants against the SFT baseline: *GRPO w/o KL*: trained with CoT prompts but without the KL penalty term. *GRPO w/o CoT*: trained without CoT prompts, using question-answering format but retaining the KL penalty.

**KL penalty is not the primary factor for preserving task-specific knowledge.** As shown in Table 3, removing the KL penalty (*GRPO w/o KL*) causes no degradation in performance on the continual learning sequence. The final average accuracy

*Table 7.* Performance evaluation on text-only tasks using Qwen2.5-7B-Instruct.

| Method | Task Order | GSM8K | USMLE | AvgAcc | FM |
|--------|------------|-------|-------|--------|-----|
| GRPO | GSM8K→USMLE | 84.2 | 62.3 | 73.3 | -1.8 |
| | USMLE→GSM8K | 85.1 | 60.7 | 72.9 | -1.2 |
| SFT | GSM8K→USMLE | 71.3 | 58.2 | 64.8 | -10.4 |
| | USMLE→GSM8K | 82.4 | 49.6 | 66.0 | -8.7 |

*Table 8.* Performance evaluation under different task orderings.

| Model | Task Order | Task 1 | Task 2 | Task 3 | AvgAcc | FM |
|-------|------------|--------|--------|--------|--------|-----|
| Qwen2.5-VL-3B | sCLEVR→SciQA→TextVQA | 57.8 | 92.7 | 72.8 | 74.4 | -0.4 |
| | TextVQA→SciQA→sCLEVR | 72.8 | 92.1 | 57.8 | 74.2 | -0.3 |
| Qwen3-VL-8B | sCLEVR→SciQA→TextVQA | 57.0 | 96.3 | 76.1 | 76.5 | -0.2 |
| | TextVQA→SciQA→sCLEVR | 76.1 | 96.8 | 55.6 | 76.2 | -0.4 |

remains, demonstrating that the KL penalty is *not* the primary mechanism preventing task-specific catastrophic forgetting. However, it is crucial to note that the training process without the KL penalty exhibits significant instability in the later stages of the task sequence. These results are obtained after multiple attempts, re-initializing from the previous task's checkpoint to achieve a convergent outcome, which suggests KL penalty primarily aids optimization stability rather than directly preventing forgetting.

**CoT is a performance booster, not a forgetting mitigator.** Our second hypothesis is also not supported by the data. The model trained without CoT (*GRPO w/o CoT*) still strongly resists forgetting, maintaining a high average accuracy across the task sequence (Table 3). In fact, it outperforms GRPO on VizWiz ( 63.8% vs. 51.8%). The general capabilities evaluation in Table 4 further confirms this conclusion. The *GRPO w/o CoT* model remains robust, and it achieves the highest score on the POPE benchmark (**88.7%**) when tested in non-CoT format evaluation. This demonstrates that while CoT can enhance performance on certain types of tasks, it is not the mechanism responsible for RFT's resistance to catastrophic forgetting. Besides, as shown in Table 3, we observe that for *GRPO w/o KL*, using CoT during inference would lead to notable hallucination.

### 5.2. Understanding RFT's Stability: Empirical Insights

Having ruled out KL penalty and CoT as primary factors, we now investigate the underlying mechanism through empirical analysis. Our key insight is that RFT exhibits selective gradient updates based on learning signal informativeness.

In policy gradient methods, the gradient for a sample $x$ is:

$$g_{\text{RFT}} = \sum_{i=1}^{n} A(a_i)\nabla_\theta \log \pi_\theta(a_i|x) \qquad (9)$$

where $A(a_i) = (r_i - \bar{r})/\sigma_r$ in GRPO. When all sampled responses receive identical rewards, $\sigma_r = 0$ and the advantage collapses. This means no parameter update occurs for samples where the model's behavior is already stable. In contrast, SFT always computes gradients toward the ground-truth response, forcing parameter updates even when the model has already converged or when the sample is far beyond the model's current capability.

**Empirical Validation.** To validate this insight, we classify training samples by the base model's success rate across rollouts:

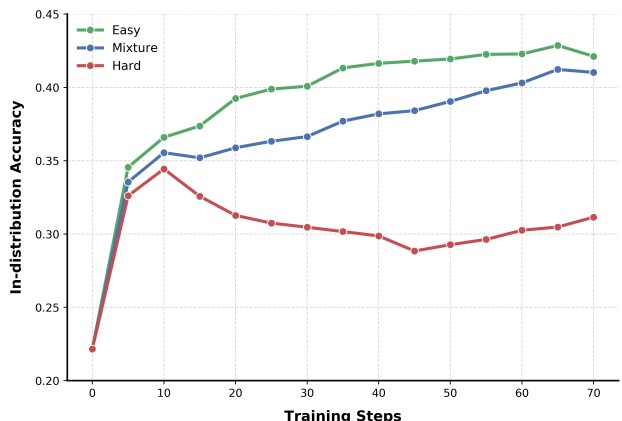

*Figure 4.* Training accuracy curves for different sample difficulty compositions. Hard samples lead to training degradation, while easy samples enable stable learning.

*easy* (at least one successful response), *hard* (zero successes), and *mixture* (random combination). Figure 4 shows training dynamics with equal sample counts. Easy samples enable stable learning through consistent reward signals. Hard samples initially improve but then degrade when no positive rewards exist, the learning signal becomes pure noise. The mixture achieves performance comparable to easy sample training, confirming that informative samples drive effective learning.

**Interpretation.** These observations suggest a coherent picture: RFT's forgetting mitigation arises from the alignment between gradient magnitude and learning signal informativeness. This prevents RFT from making arbitrary parameter changes that could interfere with previously learned knowledge. Importantly, this selective update property is inherent to policy gradient methods and does not require explicit regularization. SFT lacks this property because it pushes parameters toward ground-truth responses, regardless of the model's current state or the sample's learnability. Our explanation is empirically motivated instead of formally proven. Direct measurement of cross-task gradient interference remain important directions for future theoretical investigation.

*Table 9.* Performance and data efficiency comparison of our proposed RIF-RFT.

| Method | SciQA | TextVQA | VizWiz | GQA | Geo. | PathVQA | sCLEVR | AvgAcc | FM |
|---|---|---|---|---|---|---|---|---|---|
| SFT | 76.1 | 55.8 | 46.8 | 58.5 | 20.2 | 62.2 | 58.2 | 54.0 | -10.4 |
| GRPO | 93.0 | 74.8 | 51.8 | 65.9 | 38.4 | 41.3 | 54.2 | 60.0 | -2.3 |
| RIF-RFT | 92.9 | 73.7 | 46.6 | 63.0 | 32.3 | 40.5 | 53.7 | 57.5 | -4.5 |
| Data Kept | 81.4% | 45.6% | 42.1% | 67.6% | 37.2% | 42.5% | 52.3% | - | - |

## Algorithm 1 **R**ollout-based **I**nstance **F**iltering for **RFT** (`RIF-RFT`)

**Require:** New task training set: $\mathcal{D}_k = \{(x_i, y_i^*)\}_{i=1}^{M}$; current model policy: $\pi_\theta$; number of rollouts per input: $N$; reward threshold: $\tau$

  **Initialize** filtered dataset: $\mathcal{D}_k^{\text{filt}} \leftarrow \emptyset$

  **for** each sample $(x_i, y_i^*) \in \mathcal{D}_k$ **do**

    Initialize $R_{\text{sum}} \leftarrow 0$

    **for** $j = 1$ **to** $N$ **do**

      Sample a response: $y_{ij} \sim \pi_\theta(\cdot \mid x_i)$

      Compute reward: $R(y_{ij})$

      Update: $R_{\text{sum}} \leftarrow R_{\text{sum}} + R(y_{ij})$

    **end for**

    **if** $R_{\text{sum}}/N > \tau$ **then**

      Add $(x_i, y_i^*)$ to $\mathcal{D}_k^{\text{filt}}$

    **end if**

  **end for**

  Perform standard RFT on $\mathcal{D}_k^{\text{filt}}$ to obtain $\pi_{\theta'}$

**Ensure:** Updated model $\pi_{\theta'}$

### 5.3. RIF-RFT: Enhancing Efficiency of RFT

Our analysis in Section 5.2 shows that RFT learns most effectively from samples that provide clear, informative rewards. As illustrated in Figure 4, hard samples—those where the model fails to generate any rewarded outputs—offer almost no useful learning signal due to advantage collapse. Beyond being unhelpful, these samples introduce noise and unnecessary computational overhead. In some cases, they can even lead to subtle biases that accumulate and destabilize training. This motivates filtering such samples to improve both efficiency and stability.

To address this challenge, we propose a simple yet effective method: **R**ollout-based **I**nstance **F**iltering for **RFT** (`RIF-RFT`). The motivation is to prune the training data by discarding these incompetent samples before the RFT training. By filtering them out, RFT focuses its capacity on instances with productive learning signals, improving efficiency. Note that as training progresses, samples that initially yielded zero reward may become learnable. RIF-RFT trades this adaptability for computational savings.

The mechanism is formalized in Algorithm 1. For each instance in a new task's dataset $\mathcal{D}_k$, we perform a small number of policy rollouts. If at least one of these rollouts produces a response with a reward greater than a minimal threshold $\tau$, we classify the instance and retain it in $\mathcal{D}_k^{\text{filt}}$. As shown in Table 9, while full-data GRPO achieves the best performance in both accuracy and forgetting mitigation, RIF-RFT uses **50%** of the data while still substantially outperforming SFT in forgetting (**-4.5%** of FM). This demonstrates a practical trade-off: RIF-RFT sacrifices some per-

formance for significant computational savings, making it suitable for resource-constrained scenarios. For detailed efficiency analysis please refer to Appendix B. A natural concern is whether filtering hard samples sacrifices plasticity—the ability to learn genuinely new capabilities. Our results suggest this concern is partially mitigated: RIF-RFT's AvgAcc (**57.5%**) still substantially exceeds SFT (**54.0%**), indicating that the filtered samples contain sufficient learning signal for new tasks.

## 6. Conclusion

This work presents an investigation into the role of the fundamental learning paradigm in continual post-training for MLLMs. Our central finding is that RFT naturally mitigates the catastrophic forgetting that plagues standard SFT. Through extensive experiments, we demonstrate that while SFT leads to severe degradation of both task-specific skills and general capabilities, RFT paradigms inherently preserve that knowledge. We also introduce RIF-RFT that improves the stability and sample efficiency of RFT without compromising its robustness. Code is available at https://github.com/zhhvvv/rft_vs_sft.

## Acknowledgements

This work was supported in part by the National Natural Science Foundation of China (Grant No. 62376267), the Beijing Natural Science Foundation (Grant No. L253019), the Research Grants Council of the Hong Kong Special Administrative Region, China (GRF Project No. CityU 11217325), and the innoHK project.

We thank Dake Bu for helpful discussions. We sincerely thank the Area Chair for recognizing the contributions of our work, and we are grateful to the reviewers for their careful reading, valuable comments, and constructive suggestions, which helped us improve the paper.

## Impact Statement

This paper studies continual post-training methods for large multimodal models, with the goal of improving how such models retain previously learned capabilities while adapting to new tasks. More stable post-training can reduce the cost of repeated retraining and may improve the reliability of deployed systems. At the same time, preserving model behavior can also retain undesirable biases or unsafe capabilities, so continual post-training should be paired with careful evaluation, safety filtering, and, when necessary, targeted unlearning. Reinforcement fine-tuning is computationally expensive, and future work should continue to study more efficient training protocols and transparent reporting of compute usage.

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

In this appendix, we provide additional details to support the main findings of our study:

- **Section A:** Details on prompt templates, multimodal datasets, and evaluation benchmarks used in the experiments.

- **Section B:** Additional analyses on the computational efficiency and ablation studies of RIF-RFT.

- **Section C:** A comprehensive comparison and discussion regarding recent related studies on RFT and forgetting.

- **Section D:** A discussion on the limitations of our current approach and directions for future research.

## A. Dataset Information

**NoCoT Prompt Template**

```
Question:
<image>What is the probability that a Nile tilapia fish...
A. 2/4   B. 3/4 ... E. 4/4
You MUST provide the final answer directly.

Answer:
E
```

**CoT Prompt Template**

```
Question:
<image>What is the probability that a Nile tilapia fish...
A. 2/4   B. 3/4 ... E. 4/4
You FIRST think about the reasoning process...
The reasoning process MUST BE enclosed within
<think> </think> tags.
The final answer MUST BE put in \boxed{}.

Answer:
<think>
The Punnett square shows...
Therefore, the probability is 4/4. The correct answer is E.
</think>
\boxed{E}
```

*Figure 5.* Example prompt templates w/o and w/ CoT.

**Multimodal Datasets for Continual Post-Training and Evaluation.** Our study utilized a diverse suite of vision-language datasets for both model training and comprehensive evaluation of various multimodal capabilities, along with specialized benchmarks to assess knowledge retention and nuanced multimodal challenges. Here is a brief introduction to these datasets:

*Multimodal Datasets for Continual Post-Training:*

- **ScienceQA** (Lu et al., 2022) presents multimodal science questions requiring complex reasoning over diagrams, text, and general knowledge.

- **TextVQA** (Singh et al., 2019) focuses on questions that necessitate reading and inferring from text embedded within images.

- **VizWiz** (Gurari et al., 2018) comprises real-world image-based questions posed by visually impaired individuals, often involving ambiguity.

- **GQA** (Hudson & Manning, 2019) is designed for compositional question answering over real-world images with a strong emphasis on spatial understanding and object relationships.

- **Geometry3K** (Lu et al., 2021): This subset of MathVista (Lu et al., 2024) comprises multi-choice geometry problems equipped with dense annotations in formal language for both diagrams and text, specifically designed to evaluate complex geometric reasoning skills.

*Table 10.* Wall-clock time (hours) analysis comparing standard GRPO and RIF-RFT.

| Dataset | GRPO | RIF-RFT (Train) | RIF-RFT (Filter) | RIF-RFT |
|---------|------|-----------------|-------------------|---------|
| ScienceQA | 6.4 | 5.2 | 0.13 | 5.33 |
| TextVQA | 30.9 | 13.8 | 0.31 | 14.11 |
| VizWiz | 19.5 | 8.0 | 0.20 | 8.20 |
| GQA | 72.6 | 48.6 | 0.50 | 49.10 |
| Geometry3K | 2.3 | 0.6 | 0.02 | 0.62 |
| PathVQA | 15.4 | 5.6 | 0.15 | 5.75 |
| sCLEVR | 6.7 | 3.4 | 0.11 | 3.51 |
| Total | 153.8 | 85.2 | 1.42 | **86.62** |

- **PathVQA** (He et al., 2020) provides medical visual question answering on pathology images that demand specialized domain knowledge.

- **Super-CLEVR** (Li et al., 2023b) is a synthetic dataset crafted to rigorously test complex relational and logical reasoning.

*Benchmarks for General Knowledge Evaluation:*

- **MMMU** (Yue et al., 2024) is comprehensive benchmark comprising 11.5K college-level, multi-discipline multimodal tasks with diverse image types, demanding deliberate reasoning.

- **MMLU-Pro** (Wang et al., 2024b) is an enhanced benchmark designed for more discriminative evaluation of large language models, featuring more challenging and reasoning-focused questions with ten multiple-choice options, sourced from various academic and STEM fields.

- **POPE** (Li et al., 2023a) is a benchmark introduced to systematically investigate and assess object hallucination in vision-language large models through an improved polling-based query method.

# B. Additional Analysis

In this section, we provide additional analyses for RIF-RFT, including computational efficiency and the effect of the filtering threshold. Generalization results across model scales, text-only tasks, and task orderings are reported in the main paper.

## B.1. Computational Efficiency of RIF-RFT

Regarding the computational overhead of our proposed RIF-RFT method, we provide a detailed efficiency analysis in Table 10 on $8 \times$ H800 GPUs. The RIF-RFT process consists of a filtering phase (inference only) followed by training on the filtered data. Our analysis reveals that the filtering overhead is negligible ($<2\%$ of total time) because it avoids the costly backpropagation step. Crucially, by reducing the dataset size for the subsequent training phase, RIF-RFT achieves a $\sim 44\%$ reduction in total wall-clock time compared to standard GRPO, demonstrating that our method improves both computational and sample efficiency.

## B.2. Ablation on Filtering Threshold in RIF-RFT

In RIF-RFT, the filtering threshold $\tau$ determines which samples are retained for training. We use $\tau = 0$ as default, meaning samples are retained if they achieve any non-zero reward across rollouts. The threshold $\tau$ controls a trade-off between data quality and quantity: higher thresholds retain only samples where the model succeeds more consistently, but this reduces the volume of training data. We conduct ablation experiments on the task sequence sCLEVR $\rightarrow$ ScienceQA $\rightarrow$ TextVQA using Qwen2.5-VL-3B. Results are presented in Table 11.

*Table 11.* Ablation on filtering threshold $\tau$.

| $\tau$ | sCLEVR | SciQA | TextVQA | AvgAcc |
|--------|--------|-------|---------|--------|
| 0 | 57.2 | 92.5 | 72.1 | 73.9 |
| 0.1 | 56.8 | 92.1 | 71.4 | 73.4 |
| 0.2 | 55.9 | 91.6 | 70.2 | 72.6 |

Our default setting achieves the highest overall performance. This suggests that samples where the model has low but non-zero reward provide effective gradient signals for policy improvement. As $\tau$ increases, performance degrades across all tasks due to reduced training data volume. We recommend $\tau = 0$, which maximizes the retention of informative training instances.

### B.3. Random-Matched Baseline for RIF-RFT

To separate the effect of instance filtering from simply using fewer training samples, we compare RIF-RFT with a GRPO baseline trained on a randomly subsampled dataset of the same size as the filtered set. Table 12 shows that RIF-RFT outperforms the random-matched baseline on the PathVQA → sCLEVR subsequence, suggesting that the gain comes from selecting informative samples rather than merely reducing the amount of data.

*Table 12.* RIF-RFT and random-matched GRPO comparison on the PathVQA → sCLEVR subsequence.

| Method | PathVQA | sCLEVR | Avg | FM |
|---|---|---|---|---|
| RIF-RFT | 37.91 | 52.88 | 45.40 | -0.31 |
| RFT (Random Match) | 34.93 | 50.22 | 42.58 | -0.80 |

### B.4. POPE Split-Wise Results

POPE contains random, popular, and adversarial splits, with the adversarial split often being the most discriminative for object hallucination. To further examine hallucination behavior, we report split-wise POPE results on Qwen3-VL-8B after the continual post-training sequence. As shown in Table 13, SFT degrades performance most severely on the adversarial and random splits, while GRPO improves over the base model across all three splits.

*Table 13.* POPE split-wise results on Qwen3-VL-8B. Values in parentheses indicate changes relative to the base model.

| Method | Random | Popular | Adversarial |
|---|---|---|---|
| Base | 90.8 | 87.6 | 86.5 |
| SFT | 85.3 (↓**5.5**) | 86.9 (↓**0.7**) | 80.9 (↓**5.6**) |
| GRPO | 92.5 (↑**1.7**) | 89.7 (↑**2.1**) | 87.4 (↑**0.9**) |

## C. Comparison and discussion with concurrent works

In this section, we provide a comprehensive comparison between our study and recent concurrent works that also investigate the forgetting mitigation properties of Reinforcement Fine-Tuning.

### C.1. Comparison

Table 14 summarizes the key distinctions. While concurrent works largely attribute RFT's stability to distributional properties or data sources within single-task adaptation, our work addresses the more challenging **Continual Post-Training** (CPT) setting in Multimodal LLMs. Crucially, we offer a distinct mechanistic explanation based on gradient dynamics and sample difficulty.

### C.2. Discussion

**RL's Razor.** (Shenfeld et al., 2025) propose an empirical forgetting law, suggesting that the KL divergence between the fine-tuned and base policy on the new task is the primary predictor of forgetting. They argue that on-policy RL inherently biases updates toward KL-minimal solutions. While we agree that low KL divergence is a characteristic of RFT, our ablation studies demonstrate that removing the explicit KL penalty in GRPO does not lead to catastrophic forgetting in the CPT setting.

**Retaining by Doing.** (Chen et al., 2025) attribute RL's robustness to the use of on-policy data, arguing that the mode-seeking nature of reverse KL minimization preserves prior knowledge better than the mode-covering forward KL of SFT. Our findings complement this view but offer a more granular explanation. We show that simply using on-policy data is beneficial because it ensures the model only trains on samples within its current competence. However, our analysis of learning dynamics goes further by showing that RFT actively

*Table 14.* Comparison with concurrent studies on RFT and forgetting.

| | Ours | Shenfeld et al. (2025) | Chen et al. (2025) | Zhang et al. (2025) |
|---|---|---|---|---|
| Domain | MLLM+LLM | LLM | LLM | MLLM |
| Setting | Multiple Tasks | Single Task | Single Task | Single Task |
| Explanation | Gradient Updates | Implicit KL Minimization | On-policy Mode Seeking | Reasoning |
| KL Penalty | Not primary | Predictor of forgetting | Not primary | Not discussed |
| Method | Filtering | Oracle SFT Distribution | Iterative SFT | Reasoning SFT |

ignores uninformative samples, whereas SFT forces updates on all samples regardless of learnability. This distinction is crucial for CPT, where noise from hard samples can accumulate over sequential tasks.

**Data Perspective.** (Zhang et al., 2025) focus on the data format, showing that SFT can achieve lower forgetting if trained with reasoning trajectories (CoT) that align with the base model's perplexity. Our work demonstrates that RFT possesses this property inherently, without requiring specific data engineering or reasoning traces.

In summary, while concurrent works provide valuable insights into RFT's stability in isolated settings, our work is the first to validate these advantages in the sequential multi-task CPT setting for MLLMs and we propose a practical filtering algorithm that enhances efficiency for continual learning pipelines.

## D. Limitations and Future Work.

While our empirical analysis provides insights into RFT's stability, several limitations warrant discussion. First, due to the substantial computational cost of full-parameter fine-tuning on 7B models (each complete CPT sequence requires approximately 150 GPU-hours), we report single-run results following common practice in large-scale LLM research. Future work should verify these findings with multiple random seeds. Second, while our empirical analysis provides insights into RFT's stability, a rigorous theoretical characterization remains open. The relationship between reward signal, data source, and cross-task interference involves complex interactions that merit dedicated theoretical investigation. We hope our empirical findings motivate such future work. Third, our comparison focuses on SFT, SFT+ER, and RFT. Comparison with more sophisticated CPT methods remains an important direction for future work.

