# OpenReview forum: "Reinforcement Fine-Tuning Naturally Mitigates Forgetting in Continual Post-Training"
_ICML.cc/2026/Conference — ICML 2026 regular_

### Official Review · Reviewer_91oe · 2026-03-09

**Soundness:** 3
**Presentation:** 3
**Significance:** 2
**Originality:** 2
**Overall Recommendation:** 3
**Confidence:** 4

**Summary:**

This paper investigates the impact of supervised fine-tuning (SFT) and reinforcement fine-tuning (RFT) on catastrophic forgetting during the continual post-training of multimodal large language models (MLLMs). Through sequential multi-task experiments, the authors observe that SFT leads to severe task forgetting and degrades foundational capabilities, whereas RFT naturally preserves prior knowledge. The paper attributes this to RFT's selective update mechanism, where the advantage function collapses for hard samples, thus avoiding harmful gradient updates. Based on this empirical observation, the authors propose RIF-RFT, a data filtering algorithm that removes samples with zero initial reward to improve RFT's training efficiency.

**Compliance With Llm Reviewing Policy:**

Affirmed.

**Final Justification:**

The authors' rebuttal has partially addressed my concerns; however, I decided to keep the score due to the following reasons.

First, I think the paper seems to be overclaiming, not only in the title but throughout the paragraphs. The paper only studies the Multi-modal settings but claims for all LLM RL training without arguing the intrinsic difference or commonalities between multi-modal LLM and text-only LLM. I reckon this could be better amended at the time of submission, or the conclusions and insights of this paper may mislead a broader audience.

Second, I think the findings of this paper are not counterintuitive enough for novelty, which means that although it studies a continual setting with multiple tasks, it is not surprising that RL can be a good learner since previous literature's conclusions can extend to this. Though I appreciate the solid experiments done in this paper, sorry that nothing seems surprising to me, even from the proposed method RIF-RFT.

Overall, I may not be able to raise the score at this time, which I hope the authors can understand. In the current era of publication proliferation, it is particularly crucial for the community to uphold the standards of rigorously scoped, precise, and responsible research.

**Key Questions For Authors:**

* The core findings and mechanism analysis lack sufficient novelty. The phenomenon that RFT or online reinforcement learning mitigates forgetting is not a brand-new discovery in recent literature. Pioneering works such as "RL's Razor: Why Online Reinforcement Learning Forgets Less" and "The Path Not Taken: RLVR Provably Learns Off the Principals" have already deeply explored similar phenomena and even provided rigorous theoretical proofs. The mechanism analysis in this paper (advantage collapse on hard samples) remains relatively straightforward and empirical, failing to offer profound insights or rigorous mathematical derivations beyond existing literature.
* The design of the proposed RIF-RFT algorithm has a noticeable flaw that severely limits the model's plasticity. The algorithm employs one-time static filtering, permanently discarding "hard samples" that cannot obtain a positive reward in the initial stage. While this prevents forgetting and improves computational efficiency in the short term, it essentially confines the model to its comfort zone, completely depriving it of the opportunity to learn entirely new, complex knowledge that falls outside its current capabilities and requires long-horizon exploration. Compared to dynamic curriculum learning mechanisms, this one-size-fits-all approach appears overly heuristic.

**Limitations:**

* The core findings and mechanism analysis lack sufficient novelty. The phenomenon that RFT or online reinforcement learning mitigates forgetting is not a brand-new discovery in recent literature. Pioneering works such as "RL's Razor: Why Online Reinforcement Learning Forgets Less" and "The Path Not Taken: RLVR Provably Learns Off the Principals" have already deeply explored similar phenomena and even provided rigorous theoretical proofs. The mechanism analysis in this paper (advantage collapse on hard samples) remains relatively straightforward and empirical, failing to offer profound insights or rigorous mathematical derivations beyond existing literature.
* The design of the proposed RIF-RFT algorithm has a noticeable flaw that severely limits the model's plasticity. The algorithm employs one-time static filtering, permanently discarding "hard samples" that cannot obtain a positive reward in the initial stage. While this prevents forgetting and improves computational efficiency in the short term, it essentially confines the model to its comfort zone, completely depriving it of the opportunity to learn entirely new, complex knowledge that falls outside its current capabilities and requires long-horizon exploration. Compared to dynamic curriculum learning mechanisms, this one-size-fits-all approach appears overly heuristic.

**Strengths And Weaknesses:**

**Strengths**

* The paper targets a highly relevant and practical challenge in the continual learning of MLLMs, providing a solid amount of empirical work across various model scales and cross-modal task domains.
* The proposed RIF-RFT algorithm is remarkably simple and effective from an engineering perspective, significantly reducing the intensive computational costs of reinforcement fine-tuning and offering a highly practical solution for continuous model iteration under constrained computing resources.

**Weaknesses**

* The core findings and mechanism analysis lack sufficient novelty. The phenomenon that RFT or online reinforcement learning mitigates forgetting is not a brand-new discovery in recent literature. Pioneering works such as "RL's Razor: Why Online Reinforcement Learning Forgets Less" and "The Path Not Taken: RLVR Provably Learns Off the Principals" have already deeply explored similar phenomena and even provided rigorous theoretical proofs. The mechanism analysis in this paper (advantage collapse on hard samples) remains relatively straightforward and empirical, failing to offer profound insights or rigorous mathematical derivations beyond existing literature.
* The design of the proposed RIF-RFT algorithm has a noticeable flaw that severely limits the model's plasticity. The algorithm employs one-time static filtering, permanently discarding "hard samples" that cannot obtain a positive reward in the initial stage. While this prevents forgetting and improves computational efficiency in the short term, it essentially confines the model to its comfort zone, completely depriving it of the opportunity to learn entirely new, complex knowledge that falls outside its current capabilities and requires long-horizon exploration. Compared to dynamic curriculum learning mechanisms, this one-size-fits-all approach appears overly heuristic.
* The baseline comparisons are not convincing enough to fully demonstrate the method's superiority. When evaluating RFT's anti-forgetting capabilities, the experiments primarily compare it against basic SFT and simple experience replay (SFT+ER). It fails to include recent, advanced baselines specifically designed for continual learning in large models (such as parameter isolation architectures or LLaVA-c). The lack of direct comparison with current State-of-the-Art continual learning algorithms significantly weakens the persuasiveness of the experimental results.
* The statistical significance of the experimental results is questionable. Given the inherently high variance of reinforcement learning algorithms, the highly time-consuming continual post-training experiments in the paper are all run on a single random seed. The lack of error bar analysis from multiple independent runs is a noticeable flaw in empirical studies related to reinforcement learning, reducing the reliability of the final metric comparisons.

---

> ### Author Rebuttal · Authors · 2026-03-31
>
> We appreciate very much your constructive comments on our paper. Please kindly find our response to your comments below.
>
> ---
>
> **W1 & Q1: Novelty relative to prior RL work.**
>
> We appreciate the reviewer highlighting *RL's Razor* and *The Path Not Taken* and we have discussed the former in Section 2 and Appendix D. We will also discuss *The Path Not Taken* in revision. Both are valuable prior works studying RL's stability properties. Here, we want to respectfully argue that our contribution is distinct and non-trivial from three aspects.
>
> - First, we differ from them in settings. Existing works analyze RL's stability in *single-task* adaptation settings. Our paper addresses sequential *multi-task* continual post-training across diverse multimodal domains with cross-task interference. This distinction is important for the continual-learning setting we study. It is the gap that motivates the entire continual learning field. Single-task stability does not imply multi-task stability. In CPT, adapting to task $t$ can simultaneously degrade all tasks $1, \ldots, t{-}1$, and these interference effects compound across the full sequence. It was not obvious a priori whether the single-task stability observed in prior RL work would persist under sequential multi-task CPT.
>
> * Second, we provide a different explanation. *RL's Razor* attributes RL's stability to implicit KL minimization. We test this hypothesis in our CPT setting and find that removing the KL penalty from GRPO yields comparable downstream-task forgetting performance, although optimization becomes less stable. Our proposed explanation is a prompt-level selective-update property: when all rollout rewards for a prompt are identical, normalized advantages collapse and that prompt contributes no policy gradient.
> * We also provide several findings not covered by those prior works:
>   * RFT approaches the multi-task upper bound, which is a finding that cannot be predicted from single-task analyses.
>   * RFT not only retains task-specific knowledge but also improves general benchmarks during CPT, going beyond task-local retention.
>
>
> - We view our work as extending prior single-task insights to a more practically relevant CPT setting. We will substantially strengthen this positioning and the comparative discussion in the revision.
>
> ---
>
> **W2&Q2:** Plasticity concern of RIF-RFT.
>
> - We acknowledge that static filtering trades some plasticity for efficiency, and we do not present RIF-RFT as a universal replacement for RFT. For detailed discussion please kindly refer to our response to Reviewer rSiN (Q2).
>
> ---
>
> **W3:** Lack of more CL baselines.
>
> - We agree that stronger CPT baselines are valuable. However, we would like to clarify that our core goal is to isolate the effect of the **post-training paradigm itself**, keeping the architecture fixed without adding replay memory, task-specific experts, or routing modules. This is why we compare SFT, SFT+ER, and RFT under the same full-parameter backbone.
>
> - To contextualize RFT's performance, we reference the closest public MLLM continual-learning benchmarks [1] [2] with similar model scales (7B) and task counts. (Note: Prior works report BWT, while we report FM. Since $\text{FM}\_i = P\_{T,i} - \max\_{k \ge i} P\_{k,i}$ and $\text{BWT}\_i = P\_{T,i} - P\_{i,i}$, FM is slightly stricter, making this external comparison conservative).
>
>   | Benchmark         | Tasks | Method  | Forgetting |
>   | ----------------- | ----- | ------- | ---------- |
>   | CoIN (Table 9)    | 8     | EWC     | BWT -17.94 |
>   | CoIN (Table 9)    | 8     | LwF     | BWT -19.27 |
>   | MLLM-CL (Table 2) | 5     | DISCO   | BWT -6.46  |
>   | MLLM-CL (Table 2) | 5     | HiDe    | BWT -6.82  |
>   | MLLM-CL (Table 2) | 5     | MR-LoRA | BWT -0.01  |
>   | Ours              | 7     | SFT     | FM -10.4   |
>   | Ours              | 7     | GRPO    | FM -2.3    |
>
> - As shown, classic CL methods (EWC, LwF) still suffer severe forgetting in MLLMs. Stronger specialized methods improve this, but near-zero forgetting is typically only achieved by methods like MR-LoRA that explicitly add *task-isolated adapters and a learned router*.
>
> - By contrast, GRPO obtains FM = -2.3 over 7 tasks with **full-parameter training, no replay, and no architectural expansion**. This shows that changing SFT to RFT alone yields forgetting levels highly competitive with dedicated CL methods, while remaining orthogonal and potentially complementary to parameter-isolation approaches. We will add this positioning explicitly in the revision.
>
> ---
>
> **W4:** Statistical significance.
>
> - To directly address the variance concern, we repeat the 3-task 3B experiment over multiple independent seeds, detailed in our response to Reviewer QdsX W3.
>
> [1] CoIN: A Benchmark of Continual Instruction Tuning for Multimodal Large Language Models
>
> [2] MLLM-CL: Continual Learning for Multimodal Large Language Models

---

> > ### Author Rebuttal · Reviewer_91oe · 2026-04-04
> >
> > Thanks for the authors' rebuttal, the concerns are partially resolved. But I just wonder why the title is not about multi-modal, but the experiments are mostly of multi-modal data, is it a bit overclaiming?

---

> > > ### Author Response · Authors · 2026-04-04
> > >
> > > Thank you for raising this follow-up question. We appreciate this suggestion and are happy to refine the title to better reflect the paper's primary empirical scope.
> > >
> > > We chose this title because we want to emphasize that our observation reflects a fundamental property, not to claim that our research establishes universality across every possible continual post-training setting. We agree that the main benchmark and analysis are centered on multimodal LLM continual post-training and the text-only experiments in our paper are included as supplementary evidence to show that the trend is not only multimodal-specific.
> > >
> > > To better reflect the paper’s primary empirical scope, we are more than happy to make a minor refinement to the title to: “Reinforcement Fine-Tuning Naturally Mitigates Forgetting in **Multimodal** Continual Post-Training”. We will also adjust the abstract and introduction accordingly.
> > >
> > > We hope this addresses your remaining concern. Thank you again for helping us improve the precision of our paper.

---

### Official Review · Reviewer_QdsX · 2026-03-11

**Soundness:** 2
**Presentation:** 3
**Significance:** 3
**Originality:** 3
**Overall Recommendation:** 4
**Confidence:** 3

**Summary:**

This paper investigates the role of the learning paradigm in Continual Post-Training (CPT) for Multimodal Large Language Models (MLLMs), specifically comparing Supervised Fine-Tuning (SFT) and Reinforcement Fine-Tuning (RFT). The authors address the challenge of catastrophic forgetting, where models lose previously acquired knowledge while adapting to a stream of new tasks. The study demonstrates that while SFT leads to severe performance degradation on prior tasks and general benchmarks, RFT naturally preserves knowledge. Unlike SFT, which significantly degrades a model’s foundational abilities (base model degradation), RFT is found to protect and even enhance general model capabilities on benchmarks such as MMMU and MMLU-Pro. Through empirical analysis, the authors rule out KL penalties and Chain-of-Thought (CoT) reasoning as the primary drivers of RFT's stability . Instead, they identify that RFT’s robustness stems from a selective update mechanism: it concentrates gradient updates on samples that provide informative learning signals while ignoring "hard" samples that the model cannot yet handle, thereby preventing interference with established knowledge.

**Compliance With Llm Reviewing Policy:**

Affirmed.

**Key Questions For Authors:**

1.RIF-RFT prunes samples with zero initial reward to improve efficiency. However, if the model faces a completely new domain where all rollouts initially fail, would this filtering prevent the model from learning the task? Does this imply that RFT relies on the model having some initial competence in the target domain?

2.The paper compares RFT with vanilla SFT and ER, but not with more advanced continual learning methods such as task-specific LoRA expansion (e.g., HiDe-LLaVA), architectural decoupling (e.g., MRLoRA), or weight-consolidation approaches like Elastic Weight Consolidation. Including such comparisons would clarify whether RFT is competitive with specialized CL frameworks or mainly stronger than basic SFT baselines.

3.The paper attributes RFT’s stability to the alignment between gradient magnitude and informative learning signals, but this remains an empirical observation. Could the authors provide quantitative analysis (e.g., gradient cosine similarity) to verify whether RFT updates are more orthogonal to gradients from previous tasks than SFT updates?

**Limitations:**

The authors discuss several technical limitations in Section E, including the high computational cost that restricts experiments to single runs, the lack of a formal theoretical analysis of the selective update mechanism, and the absence of comparisons with more advanced CPT methods beyond basic SFT and experience replay. However, the discussion of potential societal impacts is minimal. The authors could further consider: (1) the environmental cost of CPT, as each full training sequence requires about 150 GPU-hours for 7B models; (2) whether RFT’s ability to preserve prior knowledge may also retain harmful biases or toxicity in the base model; and (3) safety implications, particularly whether strong resistance to forgetting could make machine unlearning or removal of sensitive information more difficult.

**Strengths And Weaknesses:**

Strengths:

1.Pioneering Paradigm Comparison Study. This paper presents the first direct comparative investigation into the impacts of Supervised Fine-Tuning (SFT) and Reinforcement Fine-Tuning (RFT) specifically within the context of Continual Post-Training (CPT) for Multimodal Large Language Models (MLLMs). It breaks the limitations of prior research that defaulted to SFT, providing a fresh perspective for the field of continual learning.

2.Revealing RFT’s Superiority in Capability Preservation. The study finds that while learning new tasks, RFT inherently mitigates catastrophic forgetting and protects—or even enhances—the base model's general capabilities, such as performance on MMMU and MMLU-Pro. This conclusion holds significant practical value for developing MLLMs that can continuously evolve without losing existing skills.

3.Profound Mechanistic Insights. Through rigorous ablation studies, the authors rule out KL penalties and Chain-of-Thought (CoT) reasoning as the primary drivers of RFT's stability and propose the "Selective Update Mechanism". This explanation, based on gradient dynamics and sample difficulty, offers deep insights into why reinforcement learning preserves knowledge more effectively.

4.Practical Efficiency Enhancement (RIF-RFT). Based on an analysis of learning dynamics, the paper introduces the RIF-RFT algorithm, which filters out samples lacking productive learning signals. This approach maintains excellent anti-forgetting performance while reducing total training time by approximately 44%, providing an efficient tool for model updates in resource-constrained environments.

Weaknesses:

1. Need for Enhanced Model Coverage. Current experiments are primarily focused on the Qwen series (Qwen2.5-VL and Qwen3-VL). To further validate the effectiveness of the RFT paradigm and the generalizability of its conclusions, it is necessary to enhance validation on other MLLMs with different architectures (e.g., LLaVA, InternVL, or other mainstream models) to rule out potential biases from a specific model family.

2. Lack of Formal Theoretical Proof. Although the paper provides a reasonable explanation for RFT’s stability through empirical experiments, such as accuracy curves for samples of varying difficulty, this explanation is largely observational and lacks rigorous mathematical characterization or formal proof. The internal connections regarding cross-task gradient interference require deeper theoretical exploration.

3. Insufficient Statistical Significance. Due to the high computational cost of full-parameter fine-tuning for 7B-scale models (approximately 150 GPU-hours per sequence), the paper primarily reports results from single runs. Without support from experiments using multiple random seeds, it is difficult to entirely rule out the possibility that subtle performance differences are coincidental.

---

> ### Author Rebuttal · Authors · 2026-03-31
>
> We thank the reviewer for the thorough evaluation. We detail our responses below point by point.
>
> ---
>
> **W1**: Need for more model coverage.
>
> - While our main experiments focus on Qwen2.5-VL-7B, we have experiments with cross-scale and cross-modality evidence. We also explain why we did not include weaker baseline models. Please see our response to Reviewer h4Fj (W2&Q1).
>
> ---
>
> **W2**: Lack of formal theoretical proof.
>
> - We want to clarify that we do not claim our analysis as a formal theorem for sequential CPT. Our explanation is grounded in a specific local property of the policy-gradient methods. We will strengthen this empirical positioning in the revision (see also our response to Reviewer rSiN W3).
>
> ---
>
> **W3**: Statistical significance of single-run experiments.
>
> - Due to the high computational cost of the full 7-task 7B CPT sequence, the main benchmark reports single runs. We repeat the 3-task 3B experiment (sCLEVR→SciQA→TextVQA subsequence) over 3 random seeds (reported as mean ± std). While this does not replace large-scale multi-seed evaluation on 7B whole task sequence setting, it strengthens the evidence that the observed effect is stable rather than cherry-picked.
>
>   *Table 1: Performance of the 3-task 3B subsequence over 3 random seeds.*
>
>   | Method |    sCLEVR |     SciQA |   TextVQA |
>   | ------ | --------: | --------: | --------: |
>   | GRPO   | 57.9±0.26 | 92.9±0.13 | 72.8±0.17 |
>   | SFT    | 51.5±0.48 | 92.1±0.36 | 67.4±0.12 |
>
> - In addition to the multiple seed runs, our claims are also supported across different algorithms and task orderings. We summarize these results here:
>
>   *Table 2: Performance comparison across RFT algorithms on the 7-task sequence.*
>
>   | Method | AvgAcc | FM    |
>   | ------ | ------ | ----- |
>   | SFT    | 54.0   | -10.4 |
>   | GRPO   | 60.0   | -2.3  |
>   | RLOO   | 59.6   | -2.1  |
>   | ReMax  | 53.9   | -3.8  |
>
>   *Table 3: Performance comparison across task orderings and modalities.*
>
>   | Order       | GRPO FM | SFT FM |
>   | ----------- | ------- | ------ |
>   | GSM8K→USMLE | -1.8    | -10.4  |
>   | USMLE→GSM8K | -1.2    | -8.7   |
>
> ---
>
> **Q1:**  RIF-RFT on completely new domains.
>
> - We agree that static filtering could be too aggressive. However, in practice, since we use well-pretrained and instruction-tuned models, this extreme case is quite rare. As shown in Table 7, the filtering still retains a substantial portion of the data.
> - If the model has zero capability on a task, we can use a warm-up SFT to gain minimal competence. This is also a standard practice in RLHF. We also want to clarify that this plasticity issue only applies to the RIF-RFT variant, and it does not change our main conclusion that RFT naturally prevents forgetting.
>
> ---
>
> **Q2**: Comparison with more advanced CL methods
>
> - Thank you for this important suggestion. Our main goal is to isolate the effect of the post-training paradigm (SFT vs. RFT) under a fixed architecture, rather than proposing new CL architectural machinery (like task-specific LoRA or routing modules).
> - Due to strict character limits in this rebuttal, we have provided a detailed discussion (including EWC, HiDe and MRLoRA as you suggested) in our response to Reviewer 91oe (W3). The data shows that simply switching from SFT to RFT achieves forgetting levels (FM = -2.3) competitive with many dedicated CL methods, without requiring any CL-specific architectural expansion. We will include this comprehensive comparison in the revised paper.
>
> ---
>
> **Q3**: Quantitative analysis of gradient interference.
>
> - To clarify, our analysis is slightly different from that of “RFT gradients are more orthogonal”. The core insight is that many prompts contribute zero or near-zero updates, whereas SFT still updates on those prompts. Therefore, our explanation does not require a change in gradient direction to reduce interference.
> - Nevertheless, we completely agree that cross-task gradient cosine analysis is useful evidence. However, obtaining meaningful cosine statistics at the 7B scale would require additional full backward passes on both current-task and previous-task batches across multiple checkpoints, which is expensive within the rebuttal window. We will include this as a concrete direction in our future work section.
>
> ---
>
> **Societal impact discussion**
>
> - We appreciate the reviewer’s suggestions and will expand the discussion on environmental cost, retention of harmful biases, and the tension between continual learning and unlearning.
>
> ---
>
> We hope these explanations address your concerns. Thank you again for your comments.

---

> > ### Author Rebuttal · Reviewer_QdsX · 2026-04-06
> >
> > The author's response resolved my concerns, and I will keep my approval.

---

### Official Review · Reviewer_h4Fj · 2026-03-13

**Soundness:** 3
**Presentation:** 3
**Significance:** 3
**Originality:** 3
**Overall Recommendation:** 5
**Confidence:** 3

**Summary:**

During continual post-training, models often suffer from catastrophic forgetting. This work compares supervised fine-tuning (SFT) and reinforcement fine-tuning (RFT) in continual learning settings, showing that SFT forgets substantially more than RFT, especially in knowledge-intensive scenarios. Beyond preserving task-specific capabilities, RFT also maintains general knowledge performance. The paper argues that this robustness primarily comes from a selective gradient update mechanism. It also introduces RIF-RFT (Rollout-based Instance Filtering for RFT), a method designed to improve the effectiveness and efficiency of RFT. Extensive experiments support these findings.

**Compliance With Llm Reviewing Policy:**

Affirmed.

**Final Justification:**

Overall, this is an interesting and well-executed paper that provides clear experimental validation of its proposed methods. Furthermore, the authors submitted a thoughtful and compelling rebuttal that effectively addressed any initial concerns. I think this paper should be accepted.

**Key Questions For Authors:**

This is a good paper with many interesting experiments.

My main question is about generalization across model families: since the reported continual post-training results are primarily based on Qwen2.5-VL, it is still unclear whether RFT’s reduction in forgetting is a general property of the algorithm or partly depends on the underlying architecture and training distribution. In particular, it would be useful to know whether the same effect holds for older and weaker models, such as LLaVA models, whose lower likelihood quality and weaker sampling capacity may change the behavior of rollout-based training.

I also have a specific question about the POPE evaluation. The paper reports using POPE as part of its general-benchmark evaluation, but POPE is typically organized into random, popular, and adversarial splits. Could you report the results by split rather than only as an aggregate score? Split-wise numbers would make the analysis more informative, especially because the adversarial split is often the most discriminative for object hallucination behavior.

**Limitations:**

yes

**Strengths And Weaknesses:**

**Strengths**:

This paper provides a systematic comparison of SFT and RFT for continual post-training, making it immediately relevant to practitioners working on knowledge-intensive tasks. The ablations on KL regularization and chain-of-thought reasoning are particularly insightful, and the experimental evaluation is extensive, covering multiple continual learning tasks and benchmarks. The selective gradient update perspective is a compelling and novel insight. The inclusion of text-only domain ablations also strengthens the empirical analysis.

**Weaknesses**:

The paper does not yet offer a strong theoretical explanation for why RFT is so effective. In addition, most experiments are conducted on a single model architecture, which limits the breadth of the conclusions. Figure 1 is also unclear and could do a better job of motivating and illustrating the core problem.

---

> ### Author Rebuttal · Authors · 2026-03-31
>
> We appreciate the positive feedback and the specific questions. Please kindly find our response to your comments below:
>
> ---
>
> **W1**: Theoretical explanation.
>
> - Our explanation focuses on an empirical mechanism instead of a rigorous theoretical proof. This offers a testable reason for the stability we observe. We will make this empirical positioning clearer in the revision. For further details, please also refer to our response to Reviewer rSiN (W3).
>
> ---
>
> **W2&Q1**: Experiments on different model families.
>
> - Currently, we focus the main study on Qwen2.5-VL-7B because it is a strong and widely used open-source backbone for multimodal RFT. Using this backbone on CPT setting makes our findings directly comparable to existing works. Nevertheless, we acknowledge  this does not fully answer cross-family generalization, and we will explicitly note this limitation in the paper.
> - Although the main benchmark uses Qwen2.5-VL-7B, we include experiments across different scales and modalities. These results show that the effect is consistent across scale and across modality, which goes beyond a single-model observation.
>
> *Table 1: Performance comparison across different model scales.*
>
> | Model Size | Method | sCLEVR | SciQA | TextVQA | AvgAcc |   FM |
> | ---------- | ------ | -----: | ----: | ------: | -----: | ---: |
> | 3B         | GRPO   |   57.8 |  92.7 |    72.8 |   74.4 | -0.4 |
> | 3B         | SFT    |   51.5 |  92.3 |    67.6 |   70.5 | -4.4 |
> | 8B         | GRPO   |   57.0 |  96.3 |    76.1 |   76.5 | -0.2 |
> | 8B         | SFT    |   48.2 |  91.5 |    68.9 |   69.5 | -7.1 |
>
> *Table 2: Performance evaluation on text-only tasks.*
>
> | Method | Task Order  | GSM8K | USMLE | AvgAcc |    FM |
> | ------ | ----------- | ----: | ----: | -----: | ----: |
> | GRPO   | GSM8K→USMLE |  84.2 |  62.3 |   73.3 |  -1.8 |
> | GRPO   | USMLE→GSM8K |  85.1 |  60.7 |   72.9 |  -1.2 |
> | SFT    | GSM8K→USMLE |  71.3 |  58.2 |   64.8 | -10.4 |
> | SFT    | USMLE→GSM8K |  82.4 |  49.6 |   66.0 |  -8.7 |
>
> - Here, we want to explain why we did not use LLaVA in the paper: In practice, current open-source RFT frameworks for GRPO training are mainly designed around Qwen-family architectures and do not natively support LLaVA. In addition, weaker baselines have lower rollout quality for on-policy training, which creates a confounding issue: insufficient rewarded samples may make forgetting effects harder to isolate. This is also reflected in the broader community trend. We will make this clear in the revision and will state cross-family validation as future work.
>
> ---
>
> **W3**: Figure 1 clarity
>
> - Thank you for pointing this out. We agree Figure 1 can be clearer and will improve the overall clarity of the figure.
>
> ---
>
> **Q2**: POPE split-wise results.
>
> - Thank you for this constructive suggestion. As an additional cross-model check on Qwen3-VL-8B, we obtain the following POPE split-wise results and observe the same pattern. We will add more split-wise results in the revision.
>
> | Method | Random      | Popular     | Adversarial |
> | :----- | :---------- | :---------- | :---------- |
> | Base   | 90.8        | 87.6        | 86.5        |
> | SFT    | 85.3 (↓5.5) | 86.9 (↓0.7) | 80.9 (↓5.6) |
> | GRPO   | 92.5 (↑1.7) | 89.7 (↑2.1) | 87.4 (↑0.9) |
>
> - These results indicate that:
>   - SFT amplifies hallucination, particularly under adversarial conditions: SFT degrades performance most severely on the Adversarial split. A comparable drop is observed on the Random split, while the Popular split remains relatively stable.
>   - Conversely, RFT improves robustness across all splits: Random (+1.7%), Popular (+2.1%), and Adversarial (+0.9%).  And the SFT–RFT gap is most pronounced on Adversarial and Random splits.
>
> ---
>
> We are grateful for your time and the constructive feedback, which has helped us significantly improve the paper.

---

> > ### Author Rebuttal · Reviewer_h4Fj · 2026-04-01
> >
> > Thank you for the clarifications. This work is timely and well executed.

---

> > > ### Author Response · Authors · 2026-04-04
> > >
> > > Thank you for confirming that your concerns have been addressed, and for recognizing the timeliness of our work. We will incorporate all the discussed improvements in the revised version.

---

### Official Review · Reviewer_rSiN · 2026-03-13

**Soundness:** 3
**Presentation:** 3
**Significance:** 4
**Originality:** 3
**Overall Recommendation:** 5
**Confidence:** 4

**Summary:**

This paper investigates forgetting in two mainstream post-training paradigms: supervised fine-tuning (SFT) and reinforcement fine-tuning (RFT). The authors report that SFT can induce catastrophic forgetting, whereas RFT tends to resist forgetting and preserves general model capabilities. They further ablate on GRPO and then attribute this stability to inherent selective gradient updates, instead of KL or Chain-of-Thought.  Building on this, they introduce a rollout-based instance filtering algorithm to improve the efficiency of RFT.

**Compliance With Llm Reviewing Policy:**

Affirmed.

**Final Justification:**

My concerns are fully addressed in the rebuttal, and I maintain my accept score. Overall, the paper addresses an important overlooked problem, provides strong and comprehensive empirical evidence, and offers a timely perspective that could influence future post-training work in both CL and RL.

**Key Questions For Authors:**

1. A natural baseline for evaluating RIF-RFT would be GRPO trained on a randomly subsampled dataset of the same size as the filtered set. This will help disentangle whether the gain comes from filtering strategy or simply less data.
2. For RIF-RFT, if a task inherently has very sparse rewards, wouldn't RIF-RFT filter out almost the entire dataset, preventing the model from learning the new task at all?
3. Figure 4: How exactly is the mixture subset sampled?
4. Figure 1 is intuitive but y-axis's label is not explicitly defined. Is it normalized to the accuracy right after that task's training?

**Limitations:**

Yes.

**Strengths And Weaknesses:**

Strengths:

(1) Clear and insightful motivation, which leads to conclusions that could be very beneficial for both CL and RLVR communities.
(2) The observation that RFT inherently acts as a CL mechanism provides a novel insight in an overcrowded literature. It is not just impactful, but offers a perspective by shifting our focus from memory, architecture, optimization to the fine-tuning paradigm itself.
(3) The empirical validation is comprehensive, covering different model scales, different domains, and different RFT algorithms. And these results are clearly organized. They also provide a repository with complete code.
(4) I appreciate the paper's structure, which makes the paper very easy to follow. The ablation study of GRPO is a standout feature. It first rules out  two obvious factors (KL and CoT),  then paves the way for a more mechanistic understanding. These experiments are designed in a very logical way.

Weaknesses:

(1) The biggest limitation is that the experiments mainly focus on the Qwen model family. Validating on architectures such as [1] [2] would strengthen the claim.
(2) The benchmarks used in this paper all have verifiable objective rewards. While this is standard for today's reasoning-focused RFT methods, it remains unclear if this forgetting mitigation holds for open-ended generation tasks.
(3) Nitpicks: There is no theoretical explanation—only extensive experimental results. The paper has some intuitive descriptions, which is sufficient for an empirical paper given the complexity of MLLMs post-training.  But a simplified formal analysis would further strengthen the paper's contribution.

[1] Liu H, Li C, Wu Q, et al. Visual instruction tuning[J]. Advances in neural information processing systems, 2023, 36: 34892-34916.
[2] Wang W, Gao Z, Gu L, et al. Internvl3. 5: Advancing open-source multimodal models in versatility, reasoning, and efficiency[J]. arXiv preprint arXiv:2508.18265, 2025.

---

> ### Author Rebuttal · Authors · 2026-03-31
>
> Thank you for your thoughtful and positive feedback on our work. We are grateful for your recognition of our research. Below are detailed explanations for remaining questions:
>
> ------
>
> **W1**: Experiments on different model architectures.
>
> - Although our main experiments are based on the Qwen architecture, our findings are not limited to a single Qwen2.5-VL-7B model. We observe the same trend across multiple models, including Qwen2.5-VL-3B, Qwen3-VL-8B, and Qwen2.5-7B.
> - We agree that validating on different model families can strengthen the claims. And we will add a discussion on this limitation in the revision. Please refer to our response to Reviewer h4Fj (W2&Q1) for more details on this point.
>
> ------
>
> **W2**: Forgetting mitigation to open-ended generation tasks.
>
> - First, we would like to emphasize that the selective update effect we observe arises from **normalized policy-gradient updates**, rather than the specific use of exact-match rewards. Whether the reward comes from exact-match verification or a learned reward model, this property holds theoretically.
> - Open-ended reward functions are typically noisier, which could introduce spurious reward variance and affect stability. We agree that validating RFT's behavior under this setting is an important future direction.
>
> ------
>
> **W3**: The lack of a formal theoretical explanation.
>
> - We appreciate this suggestion. Here, we want to clarify that our explanation is empirical rather than a complete theory, but it is grounded in a concrete and verifiable property.
> - Concretely, for GRPO on a prompt $x$, the policy gradient is $g(x)=\sum_i A_i \nabla \log \pi(a_i|x)$. When all rollout rewards for that prompt are identical, the advantages collapse, so the policy-gradient term is zero. This means that the representative policy-gradient methods we study naturally concentrate learning on prompts with informative reward variation, while SFT applies a supervised gradient on every sample.  And this explanation also provides a concrete mechanism that is distinct from prior explanations, which we have discussed in Appendix D.1. We will make this positioning clearer in the revision.
>
> ------
>
> **Q1**: Evaluating RIF-RFT against a randomly subsampled baseline.
>
> - We compare the performance of RIF-RFT against GRPO with matched data size. These results suggest that the gain is not merely an artifact of using less data. We will add there results in the revision.
>
>   *Table 1: RIF-RFT and GRPO comparison on the PathVQA→sCLEVR subsequence.*
>
>   | Method             | PathVQA | sCLEVR | Avg   | FM    |
>   | ------------------ | ------- | ------ | ----- | ----- |
>   | RIF-RFT            | 37.91   | 52.88  | 45.40 | -0.31 |
>   | RFT (Random Match) | 34.93   | 50.22  | 42.58 | -0.80 |
>
> ------
>
> **Q2**: The risk of RIF-RFT filtering out the entire dataset on novel domains.
>
> - We agree that static filtering could be too aggressive where the base model has near-zero competence. However, this extreme cases are uncommon in practice, since the base model is usually well-pretrained and instruction-tuned. In our CPT setup, the filtering still maintains a substantial portion of data (Table 7).
> - If the model has zero capability on a task, we can use a warm-up SFT to gain minimal competence. This is also a standard practice in RLHF.
> - We also want to clarify that this plasticity issue is only applied to the RIF-RFT variant. And it does not undermine our primary observation that RFT naturally prevents forgetting.
>
> ------
>
> **Q3**: Mixture subset sampling.
>
> - The mixture subset is constructed by randomly combining easy and hard samples in equal proportion, with the same total sample count as the easy-only and hard-only subsets. We will make this clear in the revised paper.
>
> ------
>
> **Q4**: Clarification of the y-axis in Figure 1.
>
> - Yes, the Performance Retention is normalized to the peak accuracy right after that task's training.

---

> > ### Author Rebuttal · Reviewer_rSiN · 2026-04-04
> >
> > Thank you for the feedback. The provided details have resolved my concerns.

---

> > > ### Author Response · Authors · 2026-04-04
> > >
> > > Thank you for acknowledging our responses. We are glad that the concerns have been resolved, and we will incorporate all discussed improvements in the revised manuscript.

---

### Decision · Program_Chairs · 2026-04-30

**Decision:**

Accept (regular)

**Comment:**

This paper investigates the impact of SFT and RL-based fine-tuning on catastrophic forgetting during the continual post-training of multimodal large language models.

The reviewers generally agree that it is a well executed study that convincingly shows that RL based fine-tuning shows significantly less forgetting effects compared to simple SFT. While there are some concerns regarding the papers focus on Qwen family models only and

Reviewer 91oe correctly points out that these results are potentially expected given recent work on forgetting after RL and SFT based fine-tuning. In this sense the paper indeed suffers from limited conceptual novelty. However, this work does provide valuable and well executed additional empirical evidence for two scenarios that have not been studied systematically: Continual learning and forgetting for CL task-sequences over multiple related tasks and for multimodal models beyond text-only LLMs.

With the proposed title change to

  Reinforcement Fine-Tuning Naturally Mitigates Forgetting in Multimodal Continual Post-Training

and an updated discussion of recent related work, this paper presents a solid and valuable contribution to the field and ICML.